# Molecular Detection and Characterization of *Borrelia garinii* (Spirochaetales: Borreliaceae) in *Ixodes nipponensis* (Ixodida: Ixodidae) Parasitizing a Dog in Korea

**DOI:** 10.3390/pathogens8040289

**Published:** 2019-12-06

**Authors:** Seung-Hun Lee, Youn-Kyoung Goo, Paul John L. Geraldino, Oh-Deog Kwon, Dongmi Kwak

**Affiliations:** 1College of Veterinary Medicine, Chungbuk National University, Cheongju 28644, Korea; dvmshlee@chungbuk.ac.kr; 2Department of Parasitology and Tropical Medicine, School of Medicine, Kyungpook National University, Daegu 41944, Korea; kuku1819@knu.ac.kr; 3Department of Biology, School of Arts and Sciences, University of San Carlos, Talamban Campus, Cebu 6000, Philippines; pjlgeraldino@usc.edu.ph; 4College of Veterinary Medicine, Kyungpook National University, Daegu 41566, Korea; odkwon@knu.ac.kr; 5Cardiovascular Research Institute, Kyungpook National University, Daegu 41566, Korea

**Keywords:** *Borrelia garinii*, dog, *Ixodes nipponensis*, Lyme borreliosis, differential diagnosis, tick-borne pathogen

## Abstract

The present study aimed to detect and characterize *Borrelia* spp. in ticks attached to dogs in Korea. Overall, 562 ticks (276 pools) attached to dogs were collected and tested for *Borrelia* infection by PCR targeting the 5S-23S rRNA intergenic spacer region (rrf-rrl). One tick larva (pool level, 0.4%; individual level, 0.2%) was confirmed by sequencing *Borrelia garinii*, a zoonotic pathogen. For molecular characterization, the outer surface protein A (*ospA*) and *flagellin* genes were analyzed. Phylogenetic *ospA* analysis distinguished *B. garinii* from *B. bavariensis*, which has been recently identified as a novel *Borrelia* species. On the other hand, phylogenetic analysis showed that single gene analysis involving rrf-rrl or *flagellin* was not sufficient to differentiate *B. garinii* from *B. bavariensis*. In addition, the *B. garinii*-infected tick was identified as *Ixodes nipponensis* by sequencing according to mitochondrial 16S rRNA and the second transcribed spacer region. To our knowledge, this is the first study to report the molecular detection of *B. garinii* in *I. nipponensis* parasitizing a dog in Korea. Continuous monitoring of tick-borne pathogens in ticks attached to animals is required to avoid disease distribution and possible transmission to humans.

## 1. Introduction

Lyme borreliosis is a tick-borne infectious zoonotic disease caused by *Borrelia burgdorferi* sensu lato (s.l.), and it involves at least 19 species. Of these species, *B. burgdorferi* sensu stricto (s.s.), *B. afzelii*, *B. garinii*, *B. bavariensis*, and *B. spielmanii* are known to be pathogenic to humans [1,2]. *B. burgdorferi* was first identified in the USA in 1982 [3], and it has been subsequently reported worldwide, including in Europe and Asia [4,5,6,7]. In the USA, 20,000 to 30,000 people are annually diagnosed with Lyme borreliosis, and in 2016, 26,000 cases were confirmed [8].

In Korea, *Borrelia* sp. was first identified in *Ixodes* ticks in 1992, and the first human case of Lyme disease was reported in 1993 [4,9]. From 2011, nationwide surveillance for Lyme disease was initiated and two cases were identified. Subsequently, it showed a gradually increasing tendency, and 23 cases were reported in 2018 [10]. Recently, different molecular and serological studies identified the nationwide distribution of the disease in humans and in various domestic animals, such as horses and dogs, and wild animals [11,12,13,14]. According to previous studies, dogs are among the most popular companion animals and are considered as sentinel animals for zoonotic diseases, including Lyme borreliosis [15,16,17,18]. It is worth noting that recent studies in Korea have shown close relationships among humans, ticks, and companion dogs with regard to tick-borne diseases [19,20].

Ticks are responsible for the transmission of various vector-borne pathogens, such as *Anaplasma* spp., *Borrelia* spp., *Ehrlichia* spp., *Rickettsia* spp., and severe fever with thrombocytopenia syndrome virus [21,22]. It is known that *B. burgdorferi* s.l. is mainly transmitted by *Ixodes* spp. [23], and consistently, *B. burgdorferi* s.l. has been isolated from *I. ricinus* and *I. persulcatus* in Korea [24]. *B. garinii* was first identified in *I. persulcatus* from vegetation in Korea in 1993, and other *Borrelia* spp., including *B. afzelii*, *B. yangtzensis*, and *B. bavariensis*, have been reported in ticks [24,25,26]. Because tick-borne diseases are associated with close relationships among humans, ticks, and companion animals, it is important to investigate tick-borne pathogens in ticks, especially those parasitizing animals, in order to avoid disease distribution and possible transmission to humans. However, to the best of our knowledge, molecular studies on *Borrelia* spp. are limited, especially in ticks parasitizing animals, in Korea.

Recently, *B. bavariensis* was separated from *B. garinii*, and it is considered a novel species. Margos et al. [27] suggested that multilocus genotying is required to classify these two *Borrelia* species. The differential diagnosis of pathogens is essential for disease diagnosis, treatment, vaccine development, and prevention. However, the suggested method is not convenient for this purpose because it requires analysis of 11 genes.

The purposes of the present study were to evaluate the prevalence of *Borrelia* spp. in ticks attached to dogs in Korea and to assess the molecular characteristics of the 5S-23S rRNA intergenic spacer region (rrf-rrl), outer surface protein A (*ospA*), and *flagellin* genes using phylogenetic analysis and determine whether single gene analysis can be used for species identification.

## 2. Results

### 2.1. Molecular Identification of Borrelia Spp.

In nested PCR, 1 of the 276 tested tick pools (pool level, 0.4%; individual level, 0.2%) was positive for the rrf-rrl fragment of *Borrelia* spp. In addition, the *ospA* and *flagellin* genes in the rrf-rrl-positive sample were amplified by PCR. All the amplified bands were single and clear. 

The *Borrelia*-positive tick was collected from a two-year-old male Alaskan Malamute in Uiseong-gun, Gyeongbuk province. The dog did not show any clinical symptoms at tick collection. Hematological and biochemical evaluations were not performed.

For species identification and molecular characterization, sequencing was performed for the three genes. On sequencing, 255, 313, and 354 bp of the rrf-rrl, *ospA*, and *flagellin* gene fragments were obtained, respectively, and all the sequences were found to be associated with *B. garinii* by the basic local alignment search tool (BLAST) search. The *B. garinii* sequences obtained in this study were deposited in the GenBank database (accession numbers: KU848760 for rrf-rrl, KU848761 for *ospA*, and MH716232 for *flagellin*).

### 2.2. Identification of Tick Species

The tick DNA sample positive for *B. garinii* was sequenced according to mitochondrial 16S rRNA and the second internal transcribed spacer region (ITS-2). On sequencing, 444 and 911 bp of the mitochondrial 16S rRNA and ITS-2 region fragments were obtained, respectively. Using BLAST and phylogenetic analysis, both sequences were found to be associated with the tick species *Ixodes nipponensis* (Figure 1 and Appendix A). The *I. nipponensis* sequences obtained in this study were deposited in the GenBank database (accession numbers: MH717250 for 16S rRNA and MH714720 for ITS-2).

### 2.3. Molecular Characterization and Phylogenetic Analysis of B. garinii

The rrf-rrl sequence identified in this study showed 98.8% (DQ150544)–97.3% (AB091797) identity with the rrf-rrl sequence of *B. garinii*. Additionally, the *ospA* sequence identified in this study showed 99.7% (AB009862)–90.4% (GU826980) identity with the *ospA* sequence of *B. garinii*. Moreover, the *flagellin* sequence identified in this study showed 99.7% (MG245785)–98.6% (KF894054) identity with the *flagellin* sequence of *B. garinii*. 

Phylogenetic analysis according to the rrf-rrl and *flagellin* genes showed that the sequences identified in this study clustered with the ones of *B. garinii*/*B. bavariensis* (Figure 2 and Figure 3). On the other hand, phylogenetic analysis according to the *ospA* gene showed that the sequence identified in this study belonged to *B. garinii*, which was different from that of *B. bavariensis* (Figure 4). All phylogenetic trees using the same data with different methods showed consistent results, that is, the major nodes were consistent among the trees, indicating that the trees constructed by the maximum parsimony method were reliable (Appendix A).

The phylogenetic trees were constructed considering the isolated host and country; however, no specific associations were noted among the molecular characteristics, host, and country. 

## 3. Discussion

*Borrelia* spp. have been identified in dogs and ticks parasitizing dogs in different countries. In the UK, 2.0% (94/4737) of ticks removed from dogs were found to be infected with *Borrelia* spp., and *B. burgdorferi* s.s., *B. garinii*, *B. afzelii*, and *B. spielmanii* were identified [17]. Additionally, ticks from dogs showed positive rates of 6.2% (13/209) in Poland, including *B. afzelii* [28], and 5.6% (6/108) in Hungary, including *B. afzelii* and *B. garinii* [29]. Serologically, dogs were found to be positive for *B. burgdorferi* s.l. at rates of 22.0% (122/555) in Australia [16], 0.9% (6/637) in China [30], 17.3% (13/75) in the Netherlands [7], and 4.7% (78/1666) in the USA [18].

*B. garinii* has been molecularly identified worldwide, including in Europe, North America, and Asia [31,32,33,34]. With regard to the isolation source, *B. garinii* has been identified in humans, animals, and ticks, including both parasitizing and nonparasitizing [31,32]. It is worth noting that there have been two clinical cases of *B. garinii* infection in dogs in Japan [33], suggesting the importance of the dog as a sentinel animal for zoonotic transmission, including *B. garinii* transmission.

In Korea, some studies have serologically identified *Borrelia* in dogs, and the seroprevalence ranges between 1.1% and 2.2% [12,35,36]; however, *Borrelia* infection in dogs has not been molecularly proven. *Haemaphysalis longicornis* is known as a dominant tick species in Korea, and other tick species, including *Ixodes*, *Amblyomma*, and *Rhipicephalus*, have been identified in different environments [37,38]. The fact that *I. nipponensis* is not a dominant tick species in dogs and in environments in Korea [38] might be the reason for the low seroprevalence of Lyme borreliosis in dogs in this country. Considering the distribution of *Borrelia* in animals, the existence of vector ticks, and the gradual increase in human clinical cases, continuous monitoring of *Borrelia* in vector ticks and animals in Korea is required.

In this study, *B. garinii* was identified in *I. nipponensis*. This result is consistent with the findings of previous studies that showed *Ixodes* as the main vector tick of *Borrelia* spp. [31]. Additionally, previous studies have experimentally confirmed *H. concinna* and *Dermacentor silvarum* as vectors of *Borrelia* spp. [39]. Consistently, *Borrelia* spp. have been identified in *Ixodes* and *Haemaphysalis* ticks in Korea and Japan [40,41]. Pal et al. [23] revealed that tick receptor for ospA (TROSPA) acts as a receptor for *Borrelia* spp. in *I. scapularis*, and the *trospa* gene was identified in *I. ricinus* and *I. persulcatus*, suggesting that these ticks could be vectors of *Borrelia* spp. [42]. Additional studies involving experimental infection are required to reveal the vector competence of ticks for *Borrelia* spp.

The rrf-rrl sequence has been widely used for the detection and differentiation of *Borrelia* spp. owing to its conserved characteristics among *Borrelia* spp. [31,40,43]. However, as suggested by De Michelis et al. [6], it is difficult to construct a reliable molecular phylogeny based on only rrf-rrl owing to the fact that the fragment is short and consists of highly conserved and variable regions. Therefore, regarding species identification, it is difficult to reliably confirm the species of *Borrelia* with single gene analysis of rrf-rrl.

It is known that *ospA* is not related to the infectivity and cause of Lyme borreliosis; however, *ospA*-deficient *Borrelia* spp. failed to colonize and survive in vector ticks [44]. In addition, a recent study found that *Borrelia* spp. showed different serotypes according to the molecular characteristics of *ospA* [45]. Another study suggested that single gene analysis is not sufficient to differentiate *B. garinii* from *B. bavariensis*, and it suggested multilocus sequence typing using housekeeping genes to differentiate *B. garinii* from *B. bavariensis* [27]. However, single phylogenetic analysis of *ospA* in this study showed the potential of *ospA* analysis for differentiating *B. garinii* from *B. bavariensis*.

*Flagellin* is a functional gene of *Borrelia* spp. and is responsible for their invasion of host cells [46]. Phylogenetic analysis of *flagellin* showed well-conserved characteristics according to species. Park et al. [47] suggested that *flagellin* could be useful for interspecies *Borrelia* differentiation, and phylogenetic analysis using the maximum parsimony (MP), maximum-likelihood (ML), and Bayesian inference (BI) methods in this study showed that *flagellin* was useful for interspecies *Borrelia* differentiation; however, it has limitations for differentiating between *B. garinii* and *B. bavariensis*.

According to the evaluation of the three selected genes using phylogenetic analysis, only the *ospA* analysis could differentiate *B. garinii* from *B. bavariensis*, including other *Borrelia* spp. On the other hand, rrf-rrl and *flagellin* gene analysis could successfully identify *Borrelia* spp., except for *B. bavariensis*. Considering that the molecular characteristics of *ospA* could induce serotype differences, which might be related to vaccine development, the differentiation of *Borrelia* spp. according to *ospA* is important. 

In conclusion, this is the first study to identify *B. garinii* in *I. nipponensis* parasitizing a dog in Korea. Phylogenetic analysis of *ospA* helped differentiate *B. garinii* from *B. bavariensis*. According to the phylogenetic analysis of *flagellin*, the *B. garinii* identified in this study showed high identity and a close relationship with other *B. garinii*, including those identified in humans. Considering the findings of previous studies on the relationship of tick-borne pathogens in dogs and humans and the increasing tendency of Lyme borreliosis in humans, continuous monitoring of tick-borne pathogens in ticks attached to animals is required to avoid disease distribution and possible transmission to humans.

## 4. Materials and Methods

### 4.1. Tick Collection and Species Identification

In this study, 562 ticks (5 larvae, 507 nymphs, and 50 adults) attached to dogs were collected from 27 regions in Gyeongbuk province, Korea between 2007 and 2015 and were preserved in 70% ethanol. The ticks were collected by practicing veterinarians at local clinics during monitoring, surveillance, and treatment or during regular check-ups after obtaining verbal consent from the dog owners. The tick collection did not require ethical approval from any authority. In addition, removal of ticks from dogs is neither harmful nor against animal welfare. In cases of larvae or nymphs, one to five tick samples were pooled depending on their sizes for DNA extraction. Finally, 276 tick pools were included in this study. 

Tick species were identified in some selected samples by sequencing according to mitochondrial 16S rRNA and ITS-2 [48,49]. Mitochondrial 16S rRNA and ITS-2 have been reported to be reliable molecular markers for tick species identification [50]. 

### 4.2. DNA Extraction, PCR, and Sequencing

DNA was extracted using the DNeasy^®^ Blood & Tissue Kit (Qiagen, Hilden, Germany) according to the manufacturer’s instructions. The quality of the extracted DNA was assessed using a spectrometer (Infinite^®^ 200 PRO NanoQuant; Tecan, Mannedorf, Switzerland).

For the detection of *Borrelia* spp., nested PCR assays were performed using the AccuPower HotStart PCR Premix Kit (Bioneer, Daejeon, Korea). The primer sets for the detection of *Borrelia* spp. were Bb23S/Bb23Sa and Bb23SnF/Bb23SanR targeting the rrf-rrl, as previously described [40]. *B. afzelii*, which was previously confirmed in our laboratory, and distilled water were used as positive and negative controls, respectively [40].

For molecular characterization, rrf-rrl-positive sample was submitted to amplify the *ospA* and *flagellin* genes, as previously described [40,51]. All the amplicons were directly sequenced by Solgent (Daejeon, Korea) bidirectionally. The obtained sequences were aligned by MUSCLE in MEGA 7.0 [52].

### 4.3. Phylogenetic Analysis

The obtained sequences were compared with those deposited in the GenBank database using the BLAST. Moreover, phylogenetic analysis was performed according to the rrf-rrl, *ospA*, and *flagellin* genes for molecular characterization. The trees were constructed using MEGA 7.0 according to the MP method with the Tree Bisection–Reconnection method for the MP search method [52].

In addition, phylogenetic trees were constructed according to the ML method (Tamura 3-parameter, gamma distributed rate) using MEGA 7.0 and the BI method (Tamura 3-parameter, gamma distributed rate) using MrBayes v3.2.6 [53]. The constructed trees were compared to confirm the absence of significant differences [54]. The best-fit model for phylogeny was selected according to the ML value, and the reliability of topology was supported by 500 bootstrap replications [52]. The sequences analyzed in this study were included with consideration of the isolated host and country.

Ticks and *Borrelia* species included in the phylogenetic analysis are summarized according to species, strain or isolate, length (bp), and GenBank accession number (Appendix A).

## Figures and Tables

**Figure 1 pathogens-08-00289-f001:**
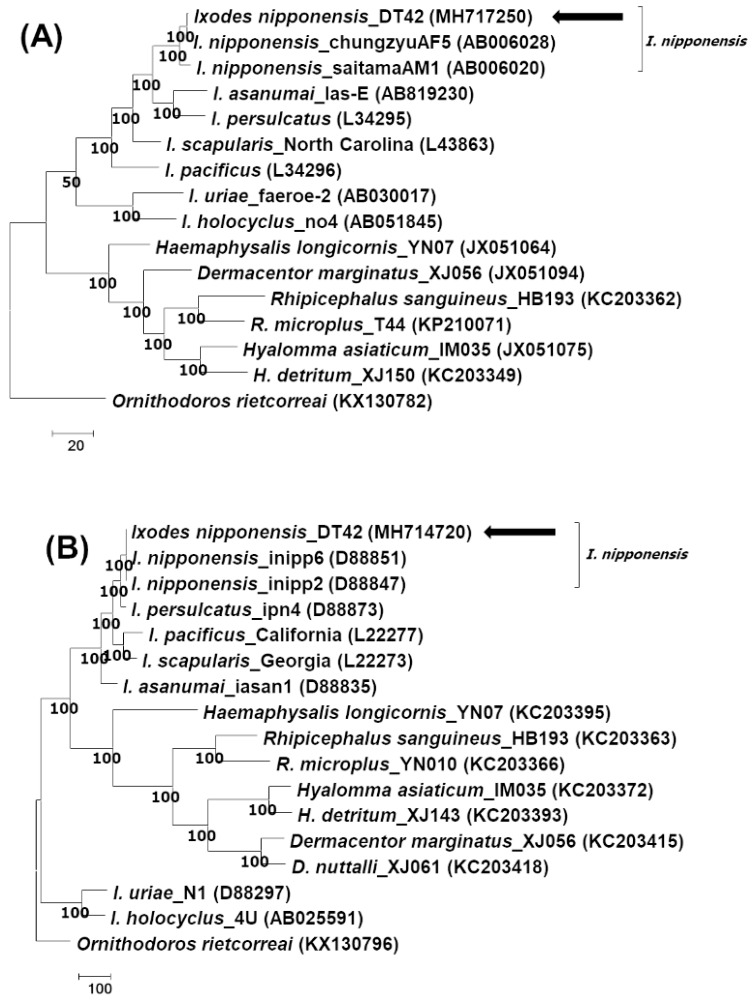
Phylogenetic analysis of the tick species. The trees are analyzed according to (**A**) mitochondrial 16S rRNA and (**B**) the second intergenic spacer region, using the maximum parsimony method by MEGA 7.0. The consensus trees inferred from the two most parsimonious trees are shown, and the cut-off value for the consensus tree is 50%. The sequences identified in this study are indicated by arrows.

**Figure 2 pathogens-08-00289-f002:**
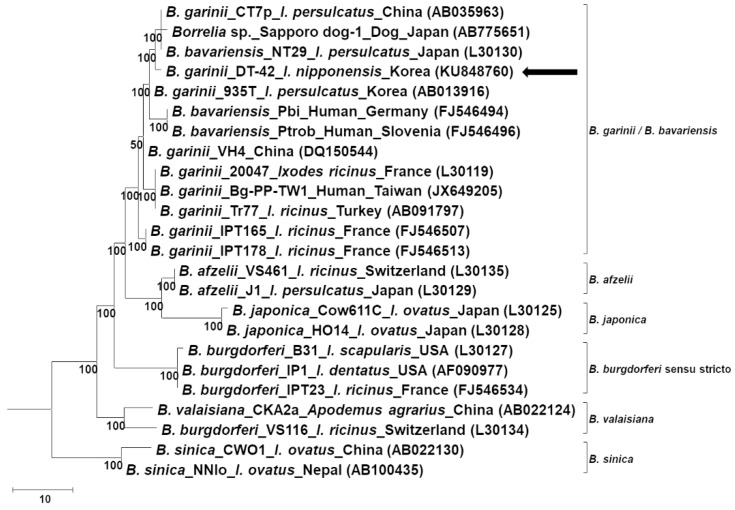
Phylogenetic analysis of *Borrelia garinii* according to the 5S–23S intergenic spacer region. The tree is constructed using the maximum parsimony method by MEGA 7.0. The consensus trees inferred from the six most parsimonious trees are shown, and the cut-off value for the consensus tree is 50%. The *Borrelia* spp., isolate, host, identified country, and nucleotide accession number are described in the tree. The sequence identified in this study is indicated by an arrow.

**Figure 3 pathogens-08-00289-f003:**
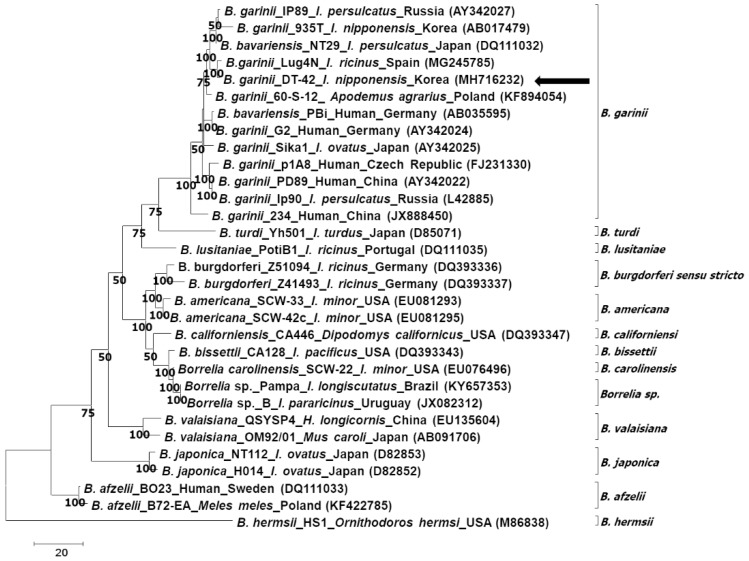
Phylogenetic analysis of *Borrelia garinii* according to the *flagellin* gene. The tree is constructed using the maximum parsimony method by MEGA 7.0. The consensus trees inferred from the eight most parsimonious trees are shown, and the cut-off value for the consensus tree is 50%. The *Borrelia* spp., isolate, host, identified country, and nucleotide accession number are described in the tree. The sequence identified in this study is indicated by an arrow.

**Figure 4 pathogens-08-00289-f004:**
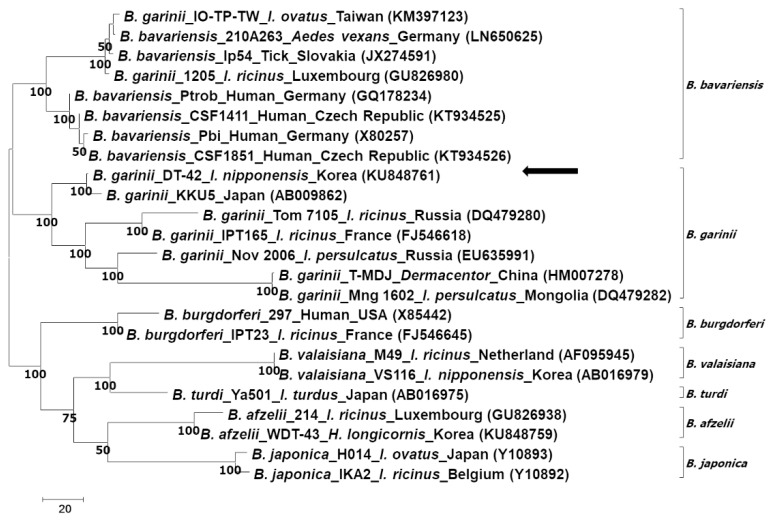
Phylogenetic analysis of *Borrelia garinii* according to the outer surface protein A (*ospA*) gene. The tree is constructed using the maximum parsimony method by MEGA 7.0. The consensus trees inferred from the four most parsimonious trees are shown, and the cut-off value for the consensus tree is 50%. The *Borrelia* spp., isolate, host, identified country, and nucleotide accession number are described in the tree. The sequence identified in this study is indicated by an arrow.

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
