# Peer review of "Molecular Detection and Characterization of *Borrelia garinii* (Spirochaetales: Borreliaceae) in *Ixodes nipponensis* (Ixodida: Ixodidae) Parasitizing a Dog in Korea"

_pathogens, 2019, doi:10.3390/pathogens8040289_

Round 1

Reviewer 1 Report

Manuscript titled as “Multilocus Genotyping of Borrelia garinii (Spirochaetales: Borreliaceae) in Ixodes nipponensis (Ixodida: Ixodidae) Parasitizing on a Dog” has done lot of phylogenetic analysis but could not presented it well even wet lab work is looks like null. Paper has some major issues, which need to be taken care off-

There is no results have been described in the result section only figures and figure legends or supplementary phylogenetic trees. What does it mean, “The positive sample also showed positivity for the ospA and flagellin genes” For every gene locus, same length of base pairs (query sequences) of other geographical isolates were used for the analysis. Why did the same phylogenetic tree was generated by two or three different methods. Is there any logic behind to do so; specially in case of Ixodes. Mention it. Result section is completely lack of any kind of information/description of using different method for the same gene. What information did you retrieve by making these many phylogenetic trees? Explain the results. Introduction and discussion is not synchronized very far away from the results where authors have used several geographical isolates of Borrelia species. Why it is necessary to do the Multilocus genotyping for species identification of this comparatively new species of parasite from their sibling species, necessity to be mention this somewhere in the beginning. In supplementary information provide the tabular form of NCBI accession ID, sequence size and other geographical details separately, rather pile up everything in the tree. Because of that, Bootstrap values are not clearly visible. Phylogenetic trees should have better quality; reconstruct them accordingly; options are available in MEGA 7.0

Overall, this manuscript needs to improvise a lot, restructure it scientifically as well as grammatically.

All the best.

Author Response

Reviewer: 1

Manuscript titled as “Multilocus Genotyping of Borrelia garinii (Spirochaetales: Borreliaceae) in Ixodes nipponensis (Ixodida: Ixodidae) Parasitizing on a Dog” has done lot of phylogenetic analysis but could not presented it well even wet lab work is looks like null. Paper has some major issues, which need to be taken care off-

Response: We appreciate the consideration of our manuscript. We have revised the manuscript according to the comments and suggestions. The revised parts in the manuscript are indicated with highlight, and our detailed responses to your comments are presented below.

There is no results have been described in the result section only figures and figure legends or supplementary phylogenetic trees.

Response: We appreciate your comment. According to your comment, we have provided additional details (host, clinical symptoms, analysis of phylogenetic trees, etc.) on page 2 (lines 73–84) and page 3 (lines 108–112).

What does it mean, “The positive sample also showed positivity for the ospA and flagellin genes”

Response: We apologize for the confusion. To improve clarity, we have revised the sentence as follows (page 2, lines 74–75): “In addition, the ospA and flagellin genes in the rrf-rrl-positive sample were amplified by PCR.”

For every gene locus, same length of base pairs (query sequences) of other geographical isolates were used for the analysis.

Response: We had assessed the “gap/missing data” option in MEGA and concluded that there was no significant difference between including and excluding gap/missing data (data not shown). Therefore, to demonstrate possible detailed differences among other sequences, we included all the gap/missing data when constructing the trees.

Why did the same phylogenetic tree was generated by two or three different methods. Is there any logic behind to do so; specially in case of Ixodes. Mention it. Result section is completely lack of any kind of information/description of using different method for the same gene. What information did you retrieve by making these many phylogenetic trees? Explain the results.

Response: There are different methods to construct phylogenetic trees, and it is known that different methods can result in different trees (results). To validate that our trees constructed using the maximum parsimony method were reliable, we additionally constructed trees using other methods, and the results showed that the major nodes among all methods were consistent. Thus, the trees were considered reliable. We have provided the required information on page 3 (lines 107–110), page 7 (lines 229–230), and page 10 (lines 373–383).

Introduction and discussion is not synchronized very far away from the results where authors have used several geographical isolates of Borrelia species. Why it is necessary to do the Multilocus genotyping for species identification of this comparatively new species of parasite from their sibling species, necessity to be mention this somewhere in the beginning.

Response: We appreciate your comment. According to your comment, we have provided information on B. garinii, B. bavariensis, and differential diagnosis on page 2 (lines 62–66) and page 6 (lines 181–186).

In supplementary information provide the tabular form of NCBI accession ID, sequence size and other geographical details separately, rather pile up everything in the tree. Because of that, Bootstrap values are not clearly visible. Phylogenetic trees should have better quality; reconstruct them accordingly; options are available in MEGA 7.0

Response: Thank you for this suggestion. We have added supplementary tables with the data of the phylogenetic trees. Please refer to Supplementary Tables S1-S5. In addition, we have revised the trees to present the data more clearly.

Overall, this manuscript needs to improvise a lot, restructure it scientifically as well as grammatically.

Response: We appreciate your comments on our manuscript. We have carefully restructured and rechecked the manuscript. In addition, this revised manuscript was professionally edited by at least two native speakers at ESSAY REVIEW company (https://essayreview.co.kr/) whose certificate was provided. Please find a revised version of our manuscript and changes were made with highlight as well.

Reviewer 2 Report

According to the authors, this study identified for the first time B. garinii in I. nipponensis parasitising on a dog in Korea. First of all, in my opinion, the country where the study took place, should be stated in the title, as similar studies from other countries are also published in the literature.

Moreover, in the Introduction, the authors present various statistics concerning borreliosis in 2016. I wonder if there are more recent numbers from the last year (2018).

In the Discussion, the authors write: "continuous monitoring of tick-borne pathogens in ticks attached to animals is required for the prevention of disease distribution and possible transmission to humans''. Based on this sentence, I would like to read why it is important to know the exact type of Borrelia.

Finally, did the dogs from where the ticks came from had any kind of symptoms;

Author Response

Reviewer: 2

According to the authors, this study identified for the first time B. garinii in I. nipponensis parasitising on a dog in Korea. First of all, in my opinion, the country where the study took place, should be stated in the title, as similar studies from other countries are also published in the literature.

Response: We appreciate the consideration of our manuscript. We have revised the manuscript according to the comments and suggestions. The revised parts in the manuscript are indicated with highlight. As suggested, we have added “in Korea” to the title.

Moreover, in the Introduction, the authors present various statistics concerning borreliosis in 2016. I wonder if there are more recent numbers from the last year (2018).

Response: We have added recent data (from 2018) on page 2 (lines 43–45).

In the Discussion, the authors write: "continuous monitoring of tick-borne pathogens in ticks attached to animals is required for the prevention of disease distribution and possible transmission to humans''. Based on this sentence, I would like to read why it is important to know the exact type of Borrelia.

Response: We appreciate your comment. The differential diagnosis of pathogens is essential for disease diagnosis, treatment, vaccine development, and prevention. In addition, according to the molecular characteristics of ospA, Borrelia can show different serotypes, which might require different vaccines. We have provided information on page 2 (lines 62–66) and page 6 (lines 181–186).

Finally, did the dogs from where the ticks came from had any kind of symptoms;

Response: Hematological or biochemical changes were not evaluated in the B. garinii-positive dog. However, clinical symptoms were not observed. We have described this on page 2 (lines 76–78).

Reviewer 3 Report

In the manuscript titled “Multilocus Geonotyping of Borrelia garinii (Spirochaetales:Borreliaceae) in Ixodes nipponensis (Ixodida:Ixodidae) Parasitising on a Dog” the authors describe the characterization of tick-borne B. garinii by multilocus genotyping. Furthermore, the authors claim that this is the first investigation reporting the molecular characterization of B. garinii in I nipponensis parasitizing a dog.

This manuscript needs improvements in the following areas.

In the last paragraph of the Introduction part, the authors describe the rationale for this study. They mention that there has been an increasing incidence of human Lyme borreliosis cases in Korea. However they do not explain the current incidence rate and the extent of increase in the disease incidence in the recent times. The existing status in Korea and how this scenario fits in at the world level should be elaborately explained. They mention that they sampled 562 ticks from dogs and they belonged to 276 pools. Furthermore, one tick larva corresponding to 0.4% pool level, 0.2% individual level was found to be positive for garinii. What is the significance of this finding in the context of disease incidence with respect to humans? How this finding might be contributing towards the increasing incidence of human Lyme borreliosis?   In the discussion part, 6th paragraph, the last sentence, the context is not clearly explained. The distinction between bavariensis as a novel species and B. garinii is not clear. How their current finding fits in this context? This should be clearly explained. What was the geographic distribution of the sampling locations within Korea? This aspect is not clear in the manuscript. How representative is this pool of samples? This aspect needs to be explained. The number of target genes used for the multilocus genotyping should be increased to make the study more relevant and thorough.

Author Response

Response to Reviewer 3:

In the manuscript titled “Multilocus Geonotyping of Borrelia garinii (Spirochaetales:Borreliaceae) in Ixodes nipponensis (Ixodida:Ixodidae) Parasitising on a Dog” the authors describe the characterization of tick-borne B. garinii by multilocus genotyping. Furthermore, the authors claim that this is the first investigation reporting the molecular characterization of B. garinii in I nipponensis parasitizing a dog.

Response: We appreciate the consideration of our manuscript. We have revised the manuscript according to the comments and suggestions. The revised parts in the manuscript are indicated with highlight, and our detailed responses to your comments are presented below.

In the last paragraph of the Introduction part, the authors describe the rationale for this study. They mention that there has been an increasing incidence of human Lyme borreliosis cases in Korea. However they do not explain the current incidence rate and the extent of increase in the disease incidence in the recent times.

Response: We appreciate the suggestion. We have revised the text and have included additional data, including recent data, on pages 1 and 2 (lines 43–45).

The existing status in Korea and how this scenario fits in at the world level should be elaborately explained.

Response: We appreciate your comment. In cases of tick-borne zoonotic diseases, including Lyme borreliosis, dogs are considered as sentinel animals. As the number of patients with Lyme borreliosis is gradually increasing in Korea, it is necessary to investigate the prevalence of Borrelia spp. in dogs and ticks parasitizing dogs. We have described this on page 2 (lines 47–50).

They mention that they sampled 562 ticks from dogs and they belonged to 276 pools. Furthermore, one tick larva corresponding to 0.4% pool level, 0.2% individual level was found to be positive for garinii. What is the significance of this finding in the context of disease incidence with respect to humans? How this finding might be contributing towards the increasing incidence of human Lyme borreliosis?  

Response: As mentioned in the previous comment, dogs are considered as sentinel animals for zoonotic diseases, including Lyme borreliosis, as dogs share many environments with their owners. In addition, in this study, on phylogenetic analysis of the flagellin gene, B. garinii showed high genetic identity with B. garinii in human cases. Currently, we cannot conclude transmission involving humans and dogs by tick biting, but our results suggest the importance of dogs and ticks parasitizing dogs in transmission. We have provided this information on page 5 (lines 142–144) and page 6 (lines 188–194).

In the discussion part, 6th paragraph, the last sentence, the context is not clearly explained. The distinction between bavariensis as a novel species and B. garinii is not clear. How their current finding fits in this context? This should be clearly explained.

Response: In the revised manuscript, the sentence has been deleted to clarify the paragraph.

What was the geographic distribution of the sampling locations within Korea? This aspect is not clear in the manuscript. How representative is this pool of samples? This aspect needs to be explained.

Response: The parasitizing ticks were collected from dogs in Gyeongbuk province, Korea. Dogs from 27 regions in Gyeongbuk province were included in this study. We have provided this information on page 6 (lines 198–200).

The number of target genes used for the multilocus genotyping should be increased to make the study more relevant and thorough.

Response: We appreciate your comment. According to your comment, we have revised the title of the manuscript as follows: “Molecular Detection and Characterization of Borrelia garinii (Spirochaetales: Borreliaceae) in Ixodes nipponensis (Ixodida: Ixodidae) Parasitizing a Dog in Korea”. In addition, single phylogenetic analysis of ospA in this study showed the potential of ospA analysis for differentiating B. garinii from B. bavariensis.

Round 2

Reviewer 1 Report

Dear Author/s,

The revised manuscript is now improved a lot from its previous version. I am happy that author has done the necessary correction in the manuscript and accept the suggestion of reviewers and did all the improvements needed for that.

Although I will suggest that please cross check one more time at your end to make the manuscript flawless.

All the best for future research.

Reviewer 3 Report

The authors have addressed all the concerns.